# Hormonal Effects on Hair Follicles

**DOI:** 10.3390/ijms21155342

**Published:** 2020-07-28

**Authors:** Monika Grymowicz, Ewa Rudnicka, Agnieszka Podfigurna, Paulina Napierala, Roman Smolarczyk, Katarzyna Smolarczyk, Blazej Meczekalski

**Affiliations:** 1Department of Gynaecological Endocrinology, Medical University of Warsaw, 4822 Warsaw, Poland; monika.grymowicz@wp.pl (M.G.); ewa.rudnicka@poczta.onet.pl (E.R.); rsmolarczyk@poczta.onet.pl (R.S.); 2Department of Gynaecological Endocrinology, Poznan University of Medical Sciences, 4861 Poznan, Poland; agnieszkapodfigurna@gmail.com (A.P.); napierala.anna92@gmail.com (P.N.); 3Department of Dermatology and Venereology, Medical University of Warsaw, 4822 Warsaw, Poland

**Keywords:** hormones, hair follicle, hair growth

## Abstract

The hair cycle and hair follicle structure are highly affected by various hormones. Androgens—such as testosterone (T); dihydrotestosterone (DHT); and their prohormones, dehydroepiandrosterone sulfate (DHEAS) and androstendione (A)—are the key factors in terminal hair growth. They act on sex-specific areas of the body, converting small, straight, fair vellus hairs into larger darker terminal hairs. They bind to intracellular androgen receptors in the dermal papilla cells of the hair follicle. The majority of hair follicles also require the intracellular enzyme 5-alpha reductase to convert testosterone into DHT. Apart from androgens, the role of other hormones is also currently being researched—e.g., estradiol can significantly alter the hair follicle growth and cycle by binding to estrogen receptors and influencing aromatase activity, which is responsible for converting androgen into estrogen (E2). Progesterone, at the level of the hair follicle, decreases the conversion of testosterone into DHT. The influence of prolactin (PRL) on hair growth has also been intensively investigated, and PRL and PRL receptors were detected in human scalp skin. Our review includes results from many analyses and provides a comprehensive up-to-date understanding of the subject of the effects of hormonal changes on the hair follicle.

## 1. Introduction

The hair cycle, as well as the structure of the hair follicle, are highly affected by various hormones. In particular, the impact of androgens has been studied and exhaustively described in previous studies [1,2]. Androgens influence hair follicles depending on the hair location on the body. The main action of androgen on the hair follicle is referred to as binding to androgen receptors in dermal papilla cells. Research studies also provide information about the locations of androgen production related to enzymes in the hair structure. Apart from androgens, the pathogenic role of other hormones is also currently being researched. In our review, the results from many analyses have been included. This review provides a comprehensive up-to-date understanding of the subject of the hormonal effects on the hair follicle. This review also highlights the significant progress in research that has been made in recent years on the effects of hormonal changes on hair at different stages of the life of women [3].

## 2. Hair Follicle Biology

### 2.1. Structure of the Hair Follicle

Hair is a filament consisting mainly of dead, keratinized cells. The hair structure consists of two parts: the hair follicle and the hair shaft (Figure 1).

The structure of the hair follicle can be divided into the upper and the lower parts. The upper part includes the infundibulum and isthmus, and the lower part is referred to as the bulb and suprabulbar region. The hair bulb is built by the dermal papilla (which includes a group of specialized fibroblasts, blood capillaries, and nerve endings) and the hair matrix (consisting of rapidly proliferating keratinocytes). The follicles of scalp hair are anchored in the subcutis, and they undergo repetitive growth cycles [4].

The hair shaft is divided into three layers: the cuticle, the cortex, and the medulla. The medulla is the outermost part of the hair which is visible above the skin. It is surrounded by a protective layer called the root sheath. The root sheath consists of two strata: the inner and the outer.

### 2.2. The Hair Follicle Cycle

The hair follicle cycle is divided into three main distinct phases: the anagen, the catagen, and the telogen (Figure 2). Some authors also identify one additional phase: the exogen.

The most prolonged phase is the anagen, which lasts 2–7 years. It is also called the growing phase. During this phase, cells divide rapidly at the lower part of the hair, while matrix cells migrate outward.

The catagen phase is a short transition period, which is defined as involution or regression. This phase lasts around three weeks. During this phase, the hair shaft loses the connections from the papillae and contracts.

The telogen phase can also be referred to as the resting stage. This phase can last about three months and is described as the regression of the matrix and retraction of the papilla to a location near the bulge. There is no significant proliferation or apoptosis during this phase.

The exogen phase (or the shedding phase) is an additional distinct phase where the active hair shaft and new hair continue to grow.

At any given time, up to 85–90% of the hair on the scalp remains in the anagen phase, whereas the remaining follicles are either in the catagen phase for 2% of the time or in the telogen phase for the remaining 10–15% of the time [5]. However, this percentage of telogen hair can be overestimated, with novel data indicating that only 3.6% remain in the telogen phase [6].

## 3. Endocrine Regulations of the Hair Follicle

### 3.1. Androgens

#### 3.1.1. General Information

Androgens are sex steroid hormones with specific locations of production. The sources of androgen are the adrenal glands, the gonads (ovaries, testes), the brain, and the placenta in pregnant women. Androgens exert their action through intracellular receptors [3]. The adrenal glands are the source of DHEA and DHEA-S, which are relatively weak androgens. The production of testosterone is limited to the testes (from puberty) in males and to the ovaries and the adrenal cortex in women of reproductive age. In females, testosterone comes in significant amounts from androstenedione. Androstenedione production takes place in equal amounts in the adrenal cortex and ovaries in females. In males, androstenedione is produced in small amounts by the testes [7].

Dihydrotestosterone is an endogenous androgen. The enzyme 5-alpha reductase is responsible for the transformation of testosterone to dihydrotestosterone at the level of tissues such as the skin, hair follicles, the prostate gland, the seminal vesicles, the liver, and the brain. Androgens can be also produced at the skin level, both de novo from cholesterol and from adrenal precursors (DHEA) [8].

#### 3.1.2. Androgen Receptor

Androgens act through an intracellular androgen receptor in the cells of the hair follicle [9]. Androgen receptors are located in human hair follicles in the dermal papilla cells. Some reports also show the expression in the outer root sheath. They are not found in the hair bulb or the bulge [10]. They are activated by the binding of androgens—testosterone and the more potent dihydrotestosterone. The main action of androgen on the hair follicle is related to binding to androgen receptors in dermal papilla cells [11]. It causes an alteration in gene expression. Insulin-like growth factor (IGF-1) is the key signal responsible for hair follicle growth stimulation. By contrast, transforming growth factor-beta (TGF-beta) is regarded as the key signal responsible for hair follicle growth inhibition [2].

#### 3.1.3. Androgen Action

The role of androgen on hair growth is undeniable. However, the influence of androgen on hair follicles depends on the hair location on the body. During puberty, an important increase in serum androgen levels is observed, and vellus hair in the pubic and axillary regions under androgen influence are transformed into terminal hairs [2]. Hair follicles in such body regions as the face, the axilla, the pubis, and the chest are subject to the stimulatory effect of androgens. The hair follicles located in the eyelashes are not under the influence of androgen. By contrast, androgens exert an inhibitory effect on the hair follicles in the region of the scalp [1]. Androgen-metabolizing enzymes can be identified in normal hair follicle locations. These enzymes play an essential role at the level of the hair follicle. Aromatase, 17 beta-HSD, and 5-alpha reductase (type I and II) are located within the outer root sheath. Aromatase and 5-alpha reductase (type I and II) are located within the inner sheath. Dermal papilla is the location of the action of enzymes such as aromatase, 17 beta-HSD, 5-alpha reductase (type II), and sulphatase. Regarding the sebaceous duct, 5-alpha reductase, and the sebaceous gland, two important enzymes can be identified: aromatase and 5-alpha reductase (type I) [1]. Despite important advances in our knowledge, there are still many aspects of the influence of androgen on the hair follicle which are unclear, and further research studies are required.

### 3.2. Sex Hormones

#### 3.2.1. Estradiol

Estradiol can significantly alter the hair follicle growth and cycle by binding to the metabolism of locally expressed high-affinity estrogen receptors. Another action of estradiol is related to its influence on the metabolism of androgen—e.g., the inhibition of aromatase activity, which is responsible for the conversion of androgen into estrogen (E2) [12].

#### 3.2.2. Progesterone

Progesterone can influence hair follicle growth through central and local action. Central action is referred to as the inhibitory effect on LH secretion, which in turn causes a decrease in ovarian theca cell stimulation (androgen synthesis). At the level of the hair follicle, progesterone decreases the conversion of testosterone to dihydrotestosterone (through the inhibition of 5-alpha reductase activity) [13].

### 3.3. Prolactin

Prolactin (PRL) is also known as luteotropin and is a polypeptide hormone encoded in the human by the PRL gene on chromosome 6 [14]. It is produced by lactotropic cells in the anterior pituitary gland, and its role is mostly associated with the growth of mammary glands during pregnancy and breast milk production. Recently, many studies concentrated on exploring new functions of prolactin, and now PRL is recognized as playing a role not only in lactation but also in reproduction, angiogenesis, osmoregulation, and hair growth [15,16,17,18]. The influence of PRL on hair growth has been intensively investigated in mammals [19,20,21]; in human scalp skin, PRL and PRL receptors (PRL-R) were identified for the first time in 2006 by Foitzik et al. The luteotropin protein was detected in a thin layer of keratinocytes, while PRL- R was detected in the outer root sheath and in the proximal inner root sheath, as well as in matrix keratinocytes. The immunoreactivity was comparable both in isolated human anagen VI and organ-cultured human hair follicles (HFs) [18]. PRL and PRL receptor immunoreactivity was also demonstrated in non-scalp skin [22], which Slominski et al. failed to prove in previous reports [23]. The very same research study identified the proinflammatory cytokines, IFNγ and TNFα, as regulators of PRL expression in HFs. Interestingly, dopamine, known as an inhibitor of PRL pituitary secretions, has no effect on PRL or PRL-R expression in human HFs [22]. The mechanism whereby prolactin directly regulates hair growth is connected with its inhibitory influence on hair shaft elongation and the premature induction of the catagen phase. Moreover, luteotropin also plays a significant role in the proliferation and apoptosis of keratinocytes in HFs by decreasing the number of Ki-67-positive cells and increasing the quantity of TUNEL + cells [18]. Prolactin is recognized as an androgen metabolism modulator. The luteotropic hormone seems to increase the level of free testosterone and dehydroepiandrosterone sulfate, decreasing at the same time the level of serum testosterone-estradiol-binding globulin [24,25]. PRL also appears to inhibit the activity of 5-alpha reductase [26].

### 3.4. Thyroid Gland Hormones

Human skin, including the hair follicles, is greatly influenced by the hypothalamic-pituitary-thyroid (HPT) axis, which controls many metabolic processes. Thyrotropin-releasing hormone (TRH, thyroliberin) secreted in the hypothalamus stimulates the pituitary gland to secrete thyroid-stimulating hormone (TSH, the thyrotropic hormone, thyrotropin). Next, TSH affects the thyroid gland to secrete thyroid hormones—thyroxine (T4) and triiodothyronine (T3). Circulating T4 acts mainly after deionizing to T3 in the peripheral organs, including human hair follicles.

Thyroid hormones play an important role in regulating the skin function which, by acting on the receptors in the cell nucleus, directly stimulates gene expression. Skin cells contain receptors for thyroid hormones; for TSH produced by the anterior pituitary gland; and even for TRH, which is formed in the hypothalamus. Thyroid hormone receptors may be in the nipple and the outer sheath hair root. The effects of thyroxine appear to be beneficial to the cell differentiation stem in keratinocytes and anagen phase extension. In animal hypothyroidism, the delayed activity of the hair follicles was observed. Triiodothyronine and thyroxine have been observed increasing melanogenesis in the hair follicles. T3 and T4 can directly affect important hair follicle functions. Some of those effects include, but are not limited to, anagen phase prolongation, the stimulation of the hair matrix, keratinocyte proliferation and pigmentation, and the modulation of intracellular keratin expression [27]. However, thyroid hormones do not significantly modulate new hair shaft formation in vitro [28]. In addition, thyroid hormones have an important role in influencing mitochondria, whose activity controls human energy metabolism and homeostasis. Mitochondria are responsible for processing fatty acid oxidation and the products of glycolysis to produce ATP.

In addition, mitochondria are involved in the metabolism of amino acids and ionic homeostasis. They possess the enzymes responsible for numerous biosynthesis processes and they deal with the regulation of cell death pathways through reactive oxygen species and Ca^2+^ signaling [29]. Due to the multifaceted and multidirectional action of mitochondria, a thorough understanding of their biological as well as clinical significance is of great importance in regulating the effect on the hair follicle [30].

Due to the fact that the hair growth process seems to be a very energy-consuming process, it is most likely that the HPT axis hormones regulate the energy of hair follicle metabolism and mitochondrial functions. Both T3 and T4 induce mitochondrial activity by stimulation exerted on keratinocytes [31].

### 3.5. Melatonin

Melatonin is synthesized mainly by the pineal gland and also in smaller amounts in the skin, the retina, the bone marrow, the digestive tract, the brain, the ovaries, and the testicles. Irrespective of its source, melatonin performs an antioxidant function, capturing and inactivating reactive oxygen species and nitrogen resulting from oxidative stress. Accurately determining the function of melatonin appears to be extremely difficult due to the complexity of the melatonin processes and its interactions, and the significant but often seemingly contradictory relationship between species and gender. It is a hormone which mainly serves to regulate the rhythm of many physiological functions. In addition, melatonin may play an important role in anti-cancer processes, in antioxidant processes, and in inhibiting cell apoptosis processes. Receptors for melatonin are found not only on the cells of the hair follicles but also on epidermal keratinocytes, dermal fibroblasts, sweat glands, and the endothelium of blood vessels. Melatonin is mainly involved in hair pigmentation by increasing the number of melanocytes, also affecting its growth, probably by stimulating the anagen phase.

It is believed that melatonin can directly and indirectly regulate hair growth. The direct effect of melatonin is based on the modulation of the serum prolactin [32]. Human hair follicles are capable of secreting melatonin, which can be stimulated by norepinephrine, as in the pineal gland. An important activity of melatonin is its ability to modulate the response of hair follicles to estrogens, which weakens the intra-follicular expression of estrogen receptors [33].

It is evident that melatonin stimulates nuclear factor erythroid-2-related factor 2, the activation of which protects the hair follicles before oxidative stress-induced hair growth inhibition [34,35]. However, understanding the role of melatonin in hair follicle biology is still very limited.

### 3.6. Other Hormones

#### 3.6.1. CRH

Corticotropin-releasing hormone (CRH) is secreted by the paraventricular nucleus of the hypothalamus. It is a central driver of the stress hormone system known as the hypothalamic-pituitary-adrenal (HPA) axis. In human skin, CRH and CRH receptors were detected for the first time almost a quarter of a century ago [36]. CRH is recognized as an inhibitor of hair shaft production, and CRH also takes part in the premature stimulation of the catagen phase. The hormone also has an influence on the reduction in the proliferation of keratinocytes in the hair matrix and the induction of their apoptosis. CRH seems to regulate hair growth both directly and indirectly—mostly through the upregulation of POMC gene expression and POMC processing in human HFs [37].

#### 3.6.2. ACTH

Adrenocorticotropic hormone (ACTH) is produced by the anterior pituitary gland, and its secretion is regulated by CRH. ACTH, on the adrenal cortex, controls the release of cortisol and androgens, in effect taking part in numerous reactions in the human body. Studies performed on murine and mink hair follicles suggest that ACTH induces the anagen phase by influencing steroid metabolism in the skin [38,39]. The role of ACTH in human hair growth is still unclear and remains to be determined.

#### 3.6.3. CORTISOL

Cortisol is a steroid hormone produced by the adrenal glands. It plays a role in a wide range of processes in the human body. However, its main function is to control the body’s stress response. Recently, particular focus has been given to hair cortisol concentration (HCC) as a promising diagnostic instrument in clinical practice [40]. Hair cortisol concentration was also reported to have an impact on correct hair growth [41]. The presence of cortisol in high levels is strictly connected with a reduction in the synthesis and premature degradation of hyaluronans and proteoglycans—important modulators of hair follicle function; however, low cortisol levels can actually bring positive effects on hair growth by slowing down the degradation of these two skin components [42].

#### 3.6.4. TRH

Thyrotropin-releasing hormone (TRH) is produced in the hypothalamus, and its principle role is to regulate the release of TSH and PRL from the anterior pituitary gland. The TRH gene is expressed in many organs of the human body, including the human hair follicle. Gáspár et al. showed that not only does TRH promote hair shaft elongation, but it also prolongs the hair cycle growth phase. Moreover, the hormone stimulates the proliferation and apoptosis of matrix keratinocytes; its effect on apoptosis seems to be connected with its influence on p53 and TGF-beta2 [43].

#### 3.6.5. GALANIN

Galanin (GAL) is a neuropeptide widely distributed throughout the central and the peripheral nervous systems. Human hair follicle has been identified as both a source and a target of GAL. This neurotransmitter is recognized as an inhibitor of human hair growth; it decreases the proliferation of matrix keratinocytes, shortens the anagen phase, and reduces hair shaft elongation [44].

## 4. Hormonal Effects on Hair at Different Stages of the Life of Women

Hormones influence hair depending on the woman’s life stage (Table 1).

### 4.1. Reproductive Age

The reproductive period in women can be affected by several hormonal disorders, such as hyperandrogenism, thyroid gland diseases, and hyperprolactinemia. These endocrine disfunctions as well as hypercortisolism and excessive growth hormone secretion, both occurring much less often, can lead to hair-growth disturbances, such as hirsutism, female pattern hair loss, and other forms of alopecia. Hirsutism is a common endocrine disorder which occurs in 5–10% of women of reproductive age, and is defined as excessive terminal hair in a male pattern in women [45]. 

Androgens, such as testosterone (T), dihydrotestosterone (DHT), and their prohormones dehydroepiandrosterone sulfate (DHEAS) and androstenedione (A) are the key factors in the growth of terminal hair. They act on sex-specific areas of the body, converting small, straight, fair vellus hairs into larger, curlier, and darker terminal hairs [46]. Hirsutism is observed in women when there is excessive growth of terminal hair in sex-specific areas, typically due to androgen excess [47]. About 70–80% of women with elevated androgens present hirsutism, although many of them manifest excessive hair without hyperandrogenemia. Hirsutism is caused mainly by an interaction between the plasma androgens and the apparent sensitivity of the hair follicle to androgen, depending on the 5-alpha reductase activity levels and subsequent binding to the androgen receptor [46,47]. To evaluate the degree of hirsutism, a modified version of the Ferriman–Gallwey visual score is used [48]. The scoring system assesses nine areas of the body and assigns a score from 0 (absence of hair) to 4 (extensive hair growth), and the maximum total score is therefore 36. In non-affected women, the total score is typically under 8 (but this depends of the ethnicity of the women). The most common cause of hirsutism is polycystic ovary syndrome (PCOS), accounting for three out of every four cases [49,50]. Other causes of androgen excess occur with much lower frequency. Nonclassic congenital adrenal hyperplasia is present in only 1.5–2.5% of women with hyperandrogenism, and androgen-secreting tumors occur in about 0.32% of these women. Diseases such as Cushing’s syndrome, hyperprolactinemia, acromegaly, and thyroid dysfunction must also be excluded as causes of androgen excess [51]. Hirsutism should be distinguished from hypertrichosis, which is excessive hair growth distributed in a generalized, nonsexual pattern and is not caused by androgen excess but is often the result of the use of certain medications (e.g., phenytoin, cyclosporine). The treatment of hirsutism is dependent on the cause and severity of the condition, but is generally based on pharmacological therapy (combined estrogen-progestin oral contraceptives, anti-androgen medication) and direct hair removal methods [51].

The second disorder associated with androgen excess in women of reproductive age is female pattern hair loss (FPHL). FPHL is characterized by a reduction in hair density in the central area of the scalp except the frontal hairline. The relationship between hair loss and androgen excess is not clear. Most women with the frontal-central pattern of hair loss have normal circulating androgens and do not present any other symptoms of hyperandrogenism, such as hirsutism or irregular periods/anovulation [52]. This type of hair loss has also been detected in women lacking an androgen receptor, with a deficiency of post-pubertal androgenization or the total absence of serum androgen. Dermatologists therefore use the phrase “female pattern hair loss” instead of androgenetic alopecia to avoid suggesting a role for androgen excess in this type of hair loss [53,54,55]. On the other hand, many women with hyperandrogenism also exhibit and complain of scalp hair loss, which indicates a role of androgens in FPHL. Enhanced androgen action in the scalp may occur due to the increased activity of 5-alpha reductase and higher concentrations of DHT or due to androgen binding to androgen receptors [53]. Female pattern hair loss may first appear in adolescence/early adulthood or in the peri- or postmenopausal age range. Patients with androgen excess usually develop FPHL during young adulthood, and the cause of FPHL in postmenopausal women is more complicated and could also be dependent on estrogen deficiency. There is also the role of genetic factors in the development of female pattern hair loss (i.e., polymorphisms in the aromatase gene) and chronic low-grade scalp inflammation [56,57,58]. The treatment of FPHL should start with minoxidil (5%), adding 5-alpha reductase inhibitors or antiandrogens when there is severe hair loss or hyperandrogenism [52]. The role and the benefit of measuring prolactin in patients with FPHL is debatable and unclear. Futterweit et al. reported in their study that among 109 patients with female pattern hair loss, 7.2% had hyperprolactinemia and 1.8% had prolactinoma [59]. After hypo- and hyperthyreosis, hyperprolactinemia is the next most common endocrine trigger of telogen effluvium. In thyrotoxicosis, the scalp hair is fine and soft and the diffuse loss of scalp hair occurs in 20–40% of patients, although the intensity of this loss is not directly related to the severity of endocrine abnormality [60]. In hyperthyreosis, alopecia areata as well as axillary, pubic, body, and eyebrow hair loss are also observed. The hair in hypothyreosis is dull, coarse, and brittle as a result of diminished sebum secretion, and diffuse alopecia is observed in up to 50% of those patients. There is also loss of genital and beard hair [60]. Many skin problems are also noticeable in patients with autoimmune thyroid disease, independently of thyroid function. Alopecia areata is typically associated with an autoimmune disorder that causes thyroid dysfunction, and diffuse alopecia can be observed in about 60% of those cases [60,61].

### 4.2. Pregnancy 

During pregnancy, the teloptosis phase is delayed and the number of shedding hairs is reduced. Moreover, the diameter of scalp hair increases during pregnancy [62]. This phenomenon is usually attributed to the effect of high levels of estrogen during gestation [62]. However, the complex changes seen in pregnancy (including increases in human chorionic gonadotropin, progesterone, prolactin, numerous growth factors, and cytokines) may well contribute to the increase in the rate of hair growth, in the hair diameter, and in the anagen/telogen ratio observed in pregnant women [63,64,65]. Hormonal changes due to gestation may cause some new terminal hair growth mainly at the abdomen, the lower back, and the thighs [66]. Sudden and severe hirsutism and/or acne during pregnancy may be a symptom of malignant ovarian or adrenal conditions, such as luteomas or Cushing’s syndrome [67,68,69].

Many patients suffer from telogen effluvium two to four months after delivery. Postpartum telogen effluvium (PPTE), a commonly described phenomenon, is explained by synchronized teloptosis and continues for 6–24 weeks and, rarely, can persist up to 15 months [70].

### 4.3. Menopause

The proportion of postmenopausal women is rising in the overall population, and issues of their general health as well as cosmetic concerns need proper attention. Female pattern hair loss (FPHL) and facial hirsutism are often observed at menopause. Marked decreases in hair density and diameter occur during the perimenopausal period or transition to menopause [71]. It is suspected that the cessation of ovarian estrogen production and complex interactions with other hormones, growth factors, and cytokines contribute to alterations in hair growth characteristics [63,71]. Estrogen levels decrease abruptly after menopause, while androgen secretion declines gradually with ageing and is maintained until the later stages of life [72]. After menopause, the increase in luteinizing hormone (LH) maintains the ovarian androgen production. In the absence of estrogens and with the tendency to accumulate visceral adipose tissue, a marked decrease in sex hormone binding globulin (SHBG) concentrations and the subsequent increase in the free androgen index are observed. Moreover, insulin resistance and hyperinsulinemia, which typically increase after menopause, may further exacerbate androgen secretion. Although the serum androgen levels do not exceed those found in premenopause, the described imbalance in estrogen and androgen production may lead to the appearance of a few terminal hairs on the face and a decrease in body and scalp hair. FPHL particularly affects the hair follicles from the parietal and frontosagittal areas. A decreased anagen phase and the regression of scalp hair to finer vellus is observed [70]. About half of the women report excessive facial growth after menopause [70]. However, understanding the role of the menopause on hair characteristics is difficult, as age-related changes may coexist and overlap with hormonal changes [63,73]. The incremental decrease in the body hair score with age suggests that this is not solely related to the endocrine changes of menopause [73]. Severe virilization with a nonmalignant origin in postmenopausal women is rare.

## 5. Conclusions

The skin can be considered to be an endocrine organ because it has been shown that it is able to synthesize a range of diverse hormones with the expression of the associated hormone receptors [74,75]. Although many hormonal paths and influences on hair growth have already been described in the literature, further studies on the full impact of hormonal regulation on hair growth need to be conducted.

## Figures and Tables

**Figure 1 ijms-21-05342-f001:**
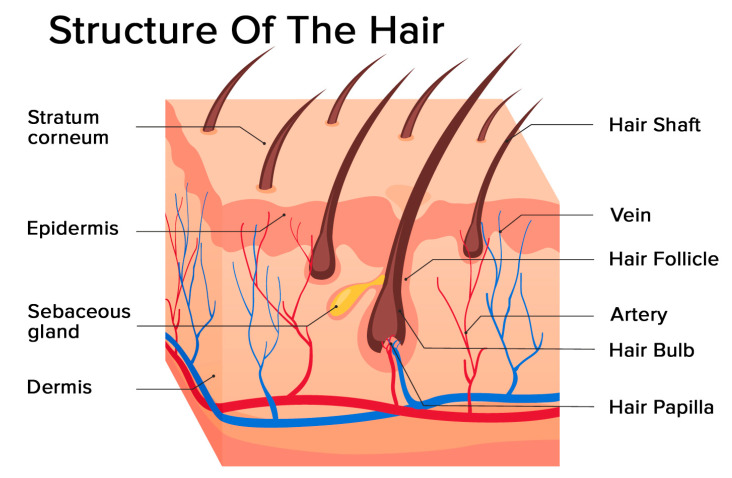
Structure of the hair: hair follicle and hair shaft (modified photo from Freepik).

**Figure 2 ijms-21-05342-f002:**
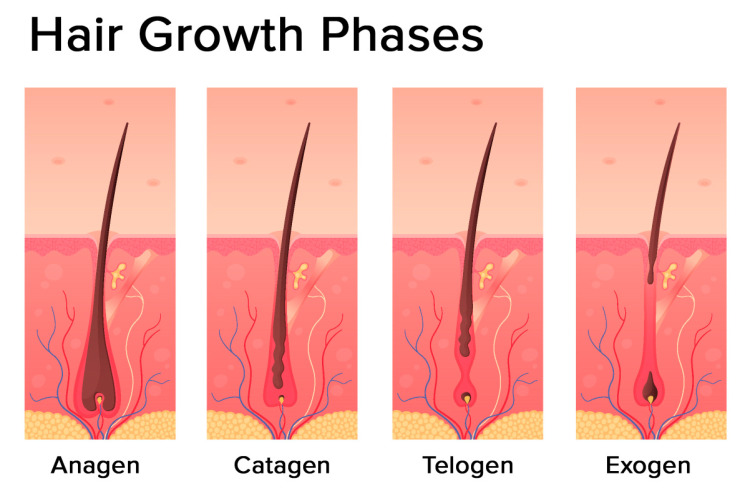
Hair growth phases: anagen (growing phase), catagen (transition phase), telogen (resting phase), exogen (shedding phase) (modified photo from Freepik).

**Table 1 ijms-21-05342-t001:** Influence of hormones on the hair cycle in different stages of female life.

Stage of Life	Main Hormones Involved	Effect
Puberty	Androgens	Transformation of vellus hair into terminal hair in the pubic and axillary regions
Reproductive age	Androgen excess (e.g PCOs, NCAH, Cushing’s syndrome, hyperprolactinemia)	Hirsutism
	Unknown/unexplained role of sex hormones	Female pattern hair loss
	Hyperthyreosis	Alopecia areata
	Hypothyreosis	Diffuse alopecia
Pregnancy	High levels of estrogen, progesterone, prolactin, and growth factors	Increases in the hair growth, in the hair diameter, and in the anagen/telogen ratio
Puerperium	Decrease in estrogen and progesterone	Postpartum telogen effluvium
Menopause	Estrogen depletion	Female pattern hair loss Facial hirsutism

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
