# Peer review of "Hormonal Effects on Hair Follicles"

_ijms, 2020, doi:10.3390/ijms21155342_

Round 1

Reviewer 1 Report

the table is very clear and it is useful for a better resume about the role

 After the revision the study is very clear. The table is useful for a better resume about the role of the hormones in different stage of life.

Reviewer 2 Report

Interesting work. Well written. Comprehensive and well presented. 

This manuscript is a resubmission of an earlier submission. The following is a list of the peer review reports and author responses from that submission.

Round 1

Reviewer 1 Report

Dear Author,

Your review is well written but given that this is a review article more papers should be reported. In addition, the are some incorrect information, e.g.:

  1. Follicles are anchored in the deeper dermal tissue by the arrector pili muscles and the sebaceous glands --> Are you referring to scalp hair follicles? If yes, they are anchored in the subcutis, and the aerecctor pili muscle and sebaceous gland have anything to do with this anchoring
  2. As expert in hair follicle biology, I have never heard "early anagen" as a separate stage. Are you referring to the difference of competent and non-competent telogen?
  3. At any given time, up to 85%–90% of the hair on the scalp remains in the anagen phase, whereas the remaining follicles are either in the catagen phase for 2% of the time or in the telogen phase for the remaining 10%–15% of the time --> This is actually incorrect, see Hernandez et al., JAAD 2018
  4. The majority of hair follicles (with the exception of pubic and axillary follicles) also require the intracellular enzyme 5-alpha reductase. --> This is not true, testosterone can trigger response, but DHT is more potent
  5. Androgen receptor is not only present in the DP, some reports also show the expression in the outer root sheath
  6. THR is a thyroid hormone, so I find very confusing that you separate this from TSH, and T3 and T4.

Reviewer 2 Report

The paper is well written.

The study reviews the relationship between androgenic hormones and hair follicle. It also assesses the effect at various stages of the woman's life.

Reviewer 3 Report

The topic is interesting and the role of various hormones in the hair cycle and hair follicle structure is well analyzed. 

The introduction is complete and the figures are clear.

For a better reading and for explain the role of androgen and other sex hormones in the hair cycle can be useful do a table with the role in a different stages of the life.